# Future Infrastructural Replacement Through the Smart Bridge Concept

**DOI:** 10.3390/ma13020405

**Published:** 2020-01-15

**Authors:** Albert D. Reitsema, Mladena Luković, Steffen Grünewald, Dick A. Hordijk

**Affiliations:** 1Engineering Structures, Delft University of Technology, Stevinweg 1, 2628 CN Delft, The Netherlands; areitsema@heijmans.nl (A.D.R.); S.Grunewald@tudelft.nl (S.G.);; 2Heijmans N.V., Graafsebaan 65, 5248 JT Rosmalen, The Netherlands; 3Department Structural Engineering and Building Materials, Ghent University, Technologiepark-Zwijnaarde 60, 9052 Ghent, Belgium; 4Adviesbureau Hageman, Polakweg 14, 2288 GG Rijswijk, The Netherlands

**Keywords:** hinder-free replacement, fast construction, low maintenance, ultra high-performance concrete (UHPC), accelerated bridge construction (ABC), advanced design

## Abstract

Most of the bridges and viaducts in the Netherlands were built in the sixties and seventies of the last century, and an increasing number of them will have to be replaced due to technical or functional reasons. The Netherlands is not an exception, many industrialized countries will face a similar replacement task in the near future. With the increased traffic intensities and the importance of mobility, the design and construction strategies for new bridges have to be different from that in the past. New methods need to ensure that traffic hindrance due to construction works and (future) maintenance activities are minimized. At the Delft University of Technology, a SMART bridge concept is being developed for fast and hindrance-free infrastructural replacement. The optimal advantage is achieved by utilizing innovative but proven technologies, and by bringing academic research into practice. A combination of recent innovations in construction technology, such as advanced cementitious materials (ACM), structural health monitoring (SHM) techniques, advanced design methods (ADM), and accelerated bridge construction (ABC) is being used. These innovations represent a step towards the next generation of infrastructure where fast construction, intelligent bridge design, sustainability, zero-energy, no/low maintenance, and aesthetics are key features.

## 1. Introduction

Bridges have always fascinated humanity. They symbolize links between people, communities, and nations [1]. Many bridges were built during the large infrastructural boom between the sixties and the eighties of the last century. The automotive industry has rapidly grown during this period, driving developed countries to construct many new bridges. An overview over the period from 1925 to 2004, which shows when bridges were built in the United States (US), Japan, Germany, and the Netherlands, is shown in Figure 1 [2,3,4,5]. 

A recent analysis [6] of data from the US Federal Highway Administration (FHWA) and the US National Bridge Inventory (NBI) indicates a strong shift in nationwide spending from building new bridges towards rehabilitation and replacement (Figure 2). Out of the 611,845 public road bridges in the US, 58,791 (9.6%) were classified as structurally deficient in 2015, and another 84,124 (13.7%) as functionally obsolete. These figures have led to claims that the US is about to experience a crisis concerning infrastructure [7]. To some extent, a similar situation is expected in many European countries. Many of the existing bridges have now reached the age of 50 years, the originally designed service life. Although this certainly does not mean that these bridges are unsafe, an increasing number will need to be replaced in the future. The replacement can be either due to deterioration or increased traffic loading, but also because of functional requirements. It is clear that, apart from rehabilitation, society will face a large infrastructural replacement task in the future.

A very common bridge type is the overpass (Figure 3). Traditionally, overpasses in the Netherlands were mainly built as cast in-situ reinforced concrete plate bridges. An advantage of a plate bridge with multiple intermediate supports compared to a single span bridge is the possibility to achieve higher slenderness due to a more favorable moment distribution. On the other hand, its main disadvantage is the increased on-site construction time needed for mounting the formwork, placing the reinforcement, and pouring and hardening of the concrete. In the past, traffic hindrance due to construction was not an issue, because these bridges were built in the open field, in the new highways. Nowadays, the Dutch highway network is one of the busiest in the world. With 3,075 km of highway in total and 73 km of highway per 1,000 km^2^ [8], the Netherlands has the highest highway density in the European Union. The busiest highways accommodate over 200,000 vehicles daily. As a result, the availability of adequate infrastructure is of the utmost importance for society. 

Traffic jams cause major problems in highly populated areas. Besides increased stress and time delays, an increase in traffic congestions may cause stalling of the economy. The cumulative combined costs of traffic congestion for the national economies of the US, United Kingdom (UK), France, and Germany are estimated to reach a staggering 4.4 trillion dollars by the year 2030 [9]. A report by the TNO [10] shows that Dutch companies lose around 1.1 billion euros annually due to traffic jams. These losses are equal to 18% of the Dutch annual infrastructure budget of around 5.9 billion euros. Therefore, one of the main demands of owners, contractors, and society in the modern construction industry is to minimize traffic hindrance. Therefore, for the next generation of road infrastructure, rapid and low hindrance bridge replacement will be a governing task.

## 2. Low Hindrance Bridge Replacement Method 

Around 60% of the highway bridges in the Netherlands are statically undetermined three- or four-span concrete plate bridges. The bridge length of the three-span bridges is between 20 m and 40 m and of four-span bridges between 40 m and 60 m [11].

When replacing these bridges, there is a desire to have maximum freedom of space for traffic lanes. This means that spanning the total length of the existing viaduct without piers in a new bridge would be very advantageous (Figure 4, indicated with red crosses). To limit hindrance and construction time, and for economic reasons, groundwork (increasing or lowering the height of both roads) should be avoided as much as possible. However, for the new bridge, the traffic profile underneath the bridge needs to be maintained (Figure 4, green box). If the existing foundations at the abutments can be (partly) reused (Figure 4, blue box), then the requirements for an innovative, new bridge replacement concept are met. 

With the replacement of a statically undetermined bridge (three- or four-spans) by a single span bridge, and the requirement to avoid altering the free space between the crossing road and the bridge, the height of the bridge deck cannot be increased, or it can, but only slightly. This requires a more slender bridge deck compared to the original one. It must be determined whether the conventional way of building satisfies this requirement and whether innovative materials and recently developed techniques play a role in solving this challenge. Though we realize that the above-sketched challenges could possibly be fully or partly addressed by steel or steel-concrete composite structures and/or alternative schemes, given that the Netherlands has great experience and tradition in precast concrete structures, the focus is on advanced, high-performance, prestressed concrete plates.

Looking at the stock of highway bridges in the Netherlands [11], a slenderness of λ = 60 (λ: span to bridge deck thickness) allows the replacement of 60% of the three-span plate bridges (Figure 5). Replacement of a concrete plate bridge of variable height by a new deck with a height equal to the minimal original deck height of the existing bridge plus an additional 100 mm or 200 mm, means that, with a slenderness of λ = 50, 52% or 95% of the bridges can be replaced, respectively [12]. Compared to the maximum slenderness of λ ≈ 30, as applied in practice with conventional (i.e., normal strength) concrete nowadays, this represents a big step. For high slenderness, but also for reasons of transportation and crane capacities, it is essential to reduce the self-weight of precast, prestressed concrete elements. Therefore, it is essential to exploit the possibilities of newly developed advanced cementitious materials (ACMs) combined with innovative building techniques.

## 3. SMART Bridge Concept

At the Delft University of Technology, a visionary concept for new infrastructure was developed, where the main focus is on bridge engineering. The total concept is called the SMART bridge [13]. A SMART bridge utilizes existing and new technologies, as well as new design and construction methods and monitoring techniques, to develop a bridge that satisfies 21st-century demands. To exploit the potential technologies, the SMART bridge concept is developed by recognizing known shortcomings in infrastructure, such as the structural or durability deficiency of existing structures. It also addresses current requirements (i.e., sustainability, low maintenance, short construction time, and no hindrance) and future possibilities (i.e., free-form design and function integration).

Opportunities have been identified to make the concept manageable and to achieve short-term utilization. During and after the process of developing a SMART bridge, newly identified societal challenges will be integrated and research will be carried out towards developing the next, more advanced version of the bridge. This approach makes the SMART bridge concept a fast-evolving innovation platform (Figure 6). 

The first SMART bridge concept combined several new technologies that incorporate new design criteria and societal demands. In the following sections, we present the experiences and future expectations of each of these technologies. We also detail steps on how to develop the first SMART bridge by application of ACMs, ABC, and the in-house developed advanced design tool (BoDeTo). Moreover, we present ways to prove its structural reliability by performing full-scale tests and by applying structural health monitoring (SHM). Attention is paid to the interactions between bridge owners, contractors, structural engineers, and building authorities (e.g., through construction tenders that incorporate societal benefits, and through considerations on how to deal with risks).

### 3.1. Advanced Cementitious Materials (ACM)

To obtain a slender bridge, a higher prestressing level has to be utilized, and therefore, ultra high-performance concrete (UHPC), a type of ACMs, must be applied. Applications of UHPC and other ACMs are still limited due to the lack of technical and economic feasibility studies, experience, design criteria, and large-scale material applications. Additionally, the costs of UHPC and its risk coverage are important factors. Since there are no design codes, the best and probably the only way to begin is to perform full-scale tests and execute pilot projects. This approach was used in Malaysia [14,15] (Figure 7a), wherein within 7 years, more than 100 road UHPC bridges were built (Figure 7b). Although in Malaysia, UHPC is mainly applied for reasons of durability and low maintenance in remote areas, the results are certainly encouraging for the application of UHPC for fast and hinder-free construction in The Netherlands.

Given its low permeability and high mechanical strength, UHPC has also attained wide applications in Switzerland, in the rehabilitation of concrete bridges in zones exposed to severe environmental conditions and mechanical loading [16,17,18]. The superior durability of UHPC over traditional concrete is an important additional benefit for new Dutch bridges since less maintenance will lead to less out of use periods, and consequently less hinder during the exploitation period.

### 3.2. Accelerated Bridge Construction (ABC)

To further decrease the total project delivery time and the downtime of the roads, the ABC method can be used. For example, the lateral bridge slide-in method or self-propelled modular transporters (SPMT’s) can be used to place a bridge (Figure 8a). The high slenderness and accompanying reduced self-weight of the bridge is crucial since the bridge needs to be lifted and hauled into position [19]. 

An example of using SPMT’s for the replacement of interstate highway bridges, which typically can take months, or even a year or more using traditional construction methods, is the project I-84 bridges over Marion Avenue in Southington, Connecticut (Figure 8b). The project consisted of the replacement of two existing bridges built in 1964. The scheduled replacement of the bridges anticipated that the highway would be closed from 9 p.m. on Friday to 5 a.m. the following Monday. However, the project progressed smoothly and both bridges were opened for traffic even before the deadline [20].

### 3.3. Advanced Design Tool (BoDeTo)

It is expected that the application of highly prestressed UHPC concrete box girders will provide opportunities for a lightweight, slender bridge concept. Despite the relatively low height of the bridge deck, a box girder shape is regarded as beneficial due to a high torsional stiffness, vast experience in the construction industry, limited need for special edge beams and full prefabrication, which leads to higher quality and faster building. We investigated various configurations of cross-sectional areas, applied prestressing force, and compressive strength of the UHPC for various spans. For each combination of these factors, the maximum achievable slenderness needed to be identified, while conforming to the requirements related to various limit states. 

A computer model named BoDeTo (Box girder Design Tool) has been developed to automate the process of hundreds of design calculations. BoDeTo is programmed with a connection between a parameter input frame (element slenderness, concrete strength, etc.), the FEM bridge design software (to determine stress distribution), and the design recommendations. As far as design codes were concerned, the maximum achievable slenderness was investigated using the Eurocode, the French recommendations for UHPC [21], and the Australian design guidelines [22]. Additionally, we investigated a configuration without shear reinforcement and with reduced concrete cover, where we did not conform to current codes in the Netherlands. Regarding dynamic response, for bridges, no additional design conditions are given in the Dutch codes. Therefore, the ratio of static deflection and first natural flexural frequency was calculated according to the New Zealand bridge design code [23]. 

Figure 9 shows the results of the calculations. For a chosen bridge span of 33 m, with a height of 0.9 m and prestressing strands of 15.7 mm, the optimum solution for the box girder cross-section and the prestressing configuration was determined. The obtained values show the results of all relevant design checks for the unity check. From the performed calculations, it could be seen that the moment capacity was governing, whereas the SLS criteria (e.g., stresses in service and deflection) were not critical. Note that, although generally being considered in the current approach (i.e., based on certain assumptions), for further development of the concept detailed attention has to be given to issues like dynamic response and the creep behavior of slender UHPC prestressed girders.

With the BoDeTo, the complete bridge design is performed within 90 seconds. Results are presented in Section 4.2.

### 3.4. Structural Health Monitoring (SHM) Techniques

Currently, SHM and SMART structures are popular terms in infrastructure research and asset management. At the moment, a lot of research is focused on different monitoring techniques such as radar, vibration monitoring, fiber optics, strain gauges, strain sensors, image analysis, and smart aggregates, among others. A common challenge with these techniques is detecting what is being measured and how to convert measured data to useful information for the assessment of the structural integrity of concrete bridges. At the Delft University of Technology, we believe that any monitoring that detects a change in the behavior of the bridge is very valuable. In the case of a new innovative bridge, for which limited experience is available and no design codes apply, SHM can contribute to confidence in the safe use of the bridge. It assures that we are in control and as soon as a change in behavior is detected, the cause of the change can be investigated, and action can be taken. An opportunity for monitoring and detecting degradation and changes in bridge behavior is through detecting changes in the stiffness of the structure. Laboratory and pilot projects [24] were carried out where the bending stiffness of concrete elements, measured with a limited number of displacement transducers, was used as an indicator for the structural health, with the final goal to develop a bridge monitoring model for the future [25]. Furthermore, it is believed that the new, fast, and contactless remote technique of radar interferometry can fruitfully be applied within the proposed concept. Radar-based measurements are discussed, for example, by Diaferio et al. [26] and Gentile and Bernardini [27].

### 3.5. Economical Most Advantageous Registration (EMVI)

In the past, the direct costs of bridge construction were key to bridge owners during the procurement phase. In bridge replacement, as presented in this paper, the direct costs of the bridge itself are not governing. The traditional methods of procurement and dealing with risk need to change to accommodate more innovative solutions. Aspects such as fast construction, sustainability, and low/no maintenance need to be considered in future tenders. Fortunately, developments in that direction are notable in the Netherlands. An example of such a tender is the project for the replacement of a three-span concrete plate bridge with a total span of 32 m of the Dutch highway A28 (Figure 10a). 

The highway crosses the underlying road N309 with three traffic lanes. The demands of the RWS (Rijkswaterstaat, the Dutch Ministry of Transportation and the Environment) were that the existing bridge should be replaced by a new bridge that will serve for the next 100 years. Additionally, the underlying road should be widened to five traffic lanes. A special feature of the project was the design and construct (D&C) contract, where a large fictive bonus, called an EMVI (Economical Most Advantageous Registration), was included for the tenderer that has the most favorable traffic model. In addition, a large bonus was available for using slender bridge girders to ensure that the underlying N309 did not need deepening to maintain the existing traffic profile under the bridge. A bonus of € 250,000 was also provided for each 50 mm in the bridge height that was reduced compared to the reference two-span bridge design (each span 24 m) consisting of 800 mm high box girders.

In the described tender procedure, the criteria for choosing the best contractor were directly related to the earned EMVI. Figure 10b shows the results of the tender for the evaluation of five contractors. The dark gray column represents the subscription price ranging from 5.7 to 6.5 (indexed). Subsequently, the fictive subscription price was calculated based on the EMVI. The white columns, ranging from 1.0 to 4.2, were the fictive subscription prices. The EMVI has a very large influence on the final result, meaning that traffic impact and infrastructure alignment were very important factors in this tender. Contractor 3 won the tender with the lowest fictive price of 1 (indexed). 

Similar tenders are expected to become the standard for future replacement tasks. Furthermore, the strategy of how to reasonably quantify certain criteria has to be reconsidered. For example, sustainability is receiving increased attention in civil engineering. Concrete is generally considered a material of which the sustainability should be improved, given that per m^3^ of concrete, 100–300 kg of CO_2_ is emitted [28]. The CO_2_ emission of UHPC per m^3^ is even higher, affecting sustainability even more unless the amount of concrete used in the structure is significantly reduced. On the other hand, a report of TNO [29] shows that the CO_2_ pollution caused by vehicles during traffic jams increases to a range of 40–70% depending on the vehicle’s weight. In 2015, the total Dutch pollution linked to road transport was 29.4 billion kg of CO_2_. Therefore, in 2015, traffic alone produced as much CO_2_ as emitted by producing concrete for around 98,000 A28/N309 viaducts, or 25 times the total Dutch highway stock. This result highlights the urgency of reducing traffic jams in modern society.

## 4. Innovative Slender and Lightweight UHPC Bridge Concept

### 4.1. Three Types of Bridges

The basic idea of the SMART bridge concept is to replace multi-span (three or four spans) viaducts with a supported bridge structure made of UHPC, spanning the total length and with minimal changes to foundations and alignments. In this way, a reduction in construction time, traffic hindrance, and full freedom in space are achieved, while keeping the same traffic profile below the bridge (Figure 4). Despite striving for a low weight of the new single-span bridge, the forces on the foundation will increase compared to the existing bridge. Therefore, a holistic approach is adopted: from the application of innovative materials and construction techniques, through investigating existing facilities in the precast concrete factory and the use of innovative tools for the structural design, to considerations related to the reuse of the existing foundations.

One solution is to develop a new very slender deck construction to achieve the required slenderness of approximately λ ≈ 50 for the bridge deck. The deck can be comprised of prestressed box girders in parallel (Figure 11, left) or it can be executed as a post-tensioned UHPC modular segmental girder bridge (Figure 11, middle). Another solution is to make use of two UHPC loadbearing girders on which the deck is hanging (Figure 11, right). Therefore, three bridge types shown in Figure 11 were examined. In the Netherlands, prestressed concrete box girders (Figure 11, left) are used widely for the construction of short- and medium-span bridges. Multiple girders are placed adjacent to each other and are connected by post-tensioning in a transverse direction, making it a fast and economical construction. As far as production of the girders is concerned, currently in the factories, the prestressing force is limited to 2250 tons (110 strands). Therefore, to achieve the required slenderness, it might be necessary to combine pretensioned prestressing with post-tensioning.

The advantage of building with prefabricated post-tensioned UHPC segments (Figure 11, middle) instead of girders is the ease of transport. On the other hand, the disadvantage is the need for temporary supports during construction. 

An optimum has to be found between bridge stiffness, prestressing force, and self-weight, considering the maximum achievable slenderness. With high slenderness, criteria such as deflections and vibrations, similarly as in steel structures, become governing. The main benefits of the system with two UHPC girders and an in-between deck (Figure 11, right) are the increased stiffness of the bridge without affecting the free traffic profile below the bridge, as well as a shift of the main load-bearing spanning from the longitudinal to the transversal direction. As a result, the required bridge deck slenderness can easily be obtained. With the two girders beside the current road profile, the width of the bridge increases, which also provides opportunities to build additional foundations adjacent to the existing ones at the abutments. Research on the application of optimization algorithms in the design of modular girders can further reduce the bridge weight [30]. The current focus of the SMART bridge project is on bridge type 1 and UHPC precast box girders, as elaborated in the following case study.

### 4.2. Case Study Replacement Task A28/N309

The existing bridge of the A28/N309 highway in the Netherlands (Figure 10a) was used for the case study. The required length of a new single-span bridge is 48 m, with a width of 16.2 m (two traffic lanes). Different challenges related to design, construction, transport, and execution were considered. Several advanced designs and/ or calculation techniques were applied. We note that:Due to the limited prestressing force in the factory for the production of prefabricated girders, a combination with post-tensioning was investigated;The lack of design codes can be overcome by up-scaled laboratory testing;The increased loading on the existing foundations can be overcome by smart adjustments in the design of the supports.

The first step was to determine the maximum achievable slenderness for the current production process and geometry of the box girders. Calculations were performed using the Eurocode (EC) (Figure 12, level 1). This meant that the concrete cover was restricted to the EC’s limits and stirrups were used to provide shear capacity. Besides shear, no traditional reinforcement was used. For the prestressing, Y1860 7 wire cables with a diameter of 15.7 mm were used. The maximum slenderness that could be reached with this box girder was λ = 40. Calculations showed that the application of concrete in a strength class that was higher than C130 did not result in an increased slenderness because of a limitation in the total allowable prestressing force of 2250 tons. 

In the second step, the pretensioned prestressing (110 strands) was combined with post-tensioning (Figure 12, Level 2). The maximum number of post-tensioning strands in the cross-section was determined using the design recommendation for Dywidag post-tensioning anchors [31]. The anchor edge and in-between distances that are given in this recommendation apply for concrete classes up to C45, resulting in a conservative calculation when using UHPC. The anchors were placed on the head ends of the box girder. For this situation, the maximum slenderness of λ = 45 could be achieved. A concrete strength over C150 does not increase the slenderness because of the limitation in the prestress force that can be applied.

The third step was to determine the achievable slenderness when the anchor distances given by Dywidag [31] were extrapolated towards values for UHPC, and when no shear reinforcement was used (Figure 12, Level 3). We assumed that for UHPC, smaller anchorage distances were applicable. As a result, a higher slenderness could be achieved in this case for concrete strength values above 150 MPa because the available space for the post-tensioning strands was no longer governing the design. The maximum slenderness that could be reached was λ = 50. In this case, fatigue becomes governing for design. Figure 13 shows the cross-section of the girder with the distribution of prestressing strands. Note that the box girder has a width of 1480 mm, which is in line with the production capabilities of the prefab company. In all calculations, the most unfavorable load combination was Load Model 1, in which a double-axle load (Tandem System) was applied in conjunction with a uniformly distributed load. More details of this case study and the analyses of the results can be found in Reference [12].

For the foundation, several calculations were performed with the intention of investigating whether it was possible to limit the force in the foundation piles to that similar to the existing bridge. In the FEM calculations, Menard’s spring stiffness’s were applied. Another option for redesigning an existing bridge foundation to increase the design capacity is by replacing a bridge girder concentrically (Figure 14b) on the abutment instead of eccentrically (Figure 14a). This removed the bending moment occurrence as a result of the abutment support reaction. This reduced the compression force in the frontal piles. 

The conclusion of the case study for replacing the existing concrete plate bridge was that a slenderness of λ = 50 could be achieved by using prefabricated C190 prestressed UHPC bridge girders. This would be achieved in combination with prestressed and post-tensioned strands, while the existing foundations at the abutments could be adapted and reused.

## 5. Discussion and Potential Impact on Society

Replacement of only one plate bridge by the SMART bridge concept already has a large impact on society because of reduced hindrance. However, as shown by using the BoDeTo tool, in the Netherlands, the potential for applying this innovative technique for existing plate bridges is very high. Provided that concrete with a very high compressive strength was used, a bridge slenderness up to 60 could be achieved (Figure 15). 

In Figure 16 (indicated by red color) it is shown that by designing with UHPC, according to the AFGC-SETRA recommendation [21], applying a reduced cover and taking into account fiber reinforcement and no stirrups, instead of requirements given by EC, the potential application area for bridge replacement increases from 16% to 58% when the height is kept equal to that of the existing plate bridge. The percentage increases from 52% to 96% when an additional height of 100 mm is available (indicated by green color). Since it concerns the application outside the codes, as done in Malaysia [14,15], prior full-scale testing would have to play a major role in demonstrating the structural capacity for short-term behavior. Meanwhile, monitoring, as applied in Switzerland [17], would contribute to the control of long-term structural behavior.

## 6. Conclusions

For the new generation of bridges, a paradigm shift will need to occur in bridge engineering. Especially in highly populated areas, continuous availability of infrastructure will have the highest priority and should be considered for upcoming large replacement tasks. Downtime caused by maintenance and construction works should be minimized as it has significant social and economic effects. In this paper, we presented a total concept and a holistic approach for the replacement of multi-span overpasses, of which there are many in the Netherlands and worldwide. The benefits of using the so-called SMART bridge concept for future replacement tasks are numerous, including:Significant reduction in traffic hindrance during construction (e.g., the 22 weeks, as is common for replacing a standard concrete plate bridge, is expected to be reduced to days).Full freedom in space under the bridge and no need to rebuild intermediate supports.No need to realign the road or reduce the available traffic profile under the bridge.By reusing existing foundations, an economical and fast building method is achieved.The enhanced durability properties of UHPC are expected to reduce maintenance needs.By reusing existing foundations, a sustainable solution is developed by applying durable UHPC girders and reducing traffic hindrance.

The proposed concept is innovative in many aspects. Innovation, apart from solving technical challenges, is also related to the implementation of these ideas and building awareness to sustain them. Whereas governments in the past believed that innovation has to be done by industry, given societal demands, cooperation is necessary between the government, industry, scientific institutions, and possibly other parties such as standardization organizations. Besides the costs of research efforts, acceptance criteria (no codes), and dealing with the risks related to unknown techniques, the non-financial aspects like a reduction of hindrance and improvement of sustainability play an important role. To be prepared for societal demands in the future concerning infrastructure, organizational and technical developments have to start now. For upcoming infrastructural replacement tasks, there is no time to lose.

## Figures and Tables

**Figure 1 materials-13-00405-f001:**
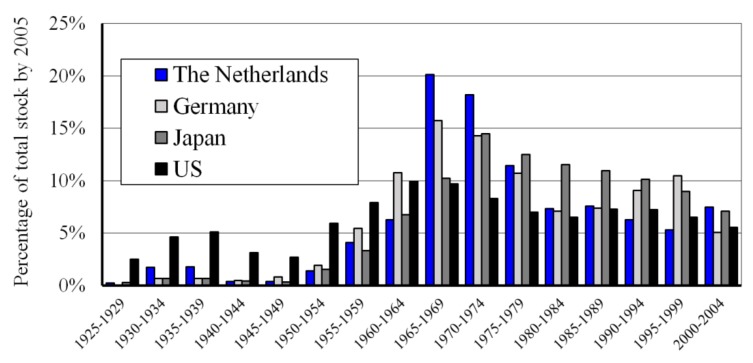
Historical overview of built bridges in Japan [2], the US [3], Germany [4], and the Netherlands [5].

**Figure 2 materials-13-00405-f002:**
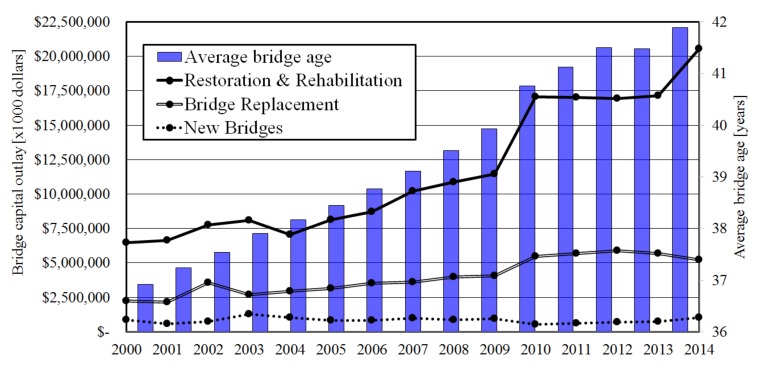
US bridges capital outlay and age (routine maintenance costs not included) [6].

**Figure 3 materials-13-00405-f003:**
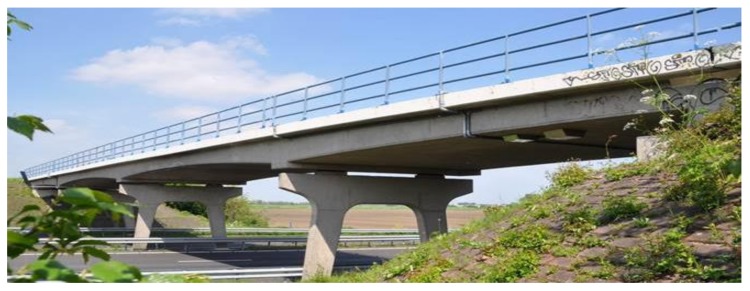
Example of a common four-span plate bridge type in the Netherlands: Bridge Zijlweg of the highway A59 in the province Brabant.

**Figure 4 materials-13-00405-f004:**
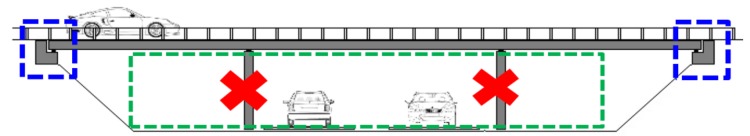
Requirements for an innovative, new bridge replacement concept for existing in-situ cast multi-span concrete plate bridges: no piers (red crosses), maintain profile (green box), no groundwork (green and blue boxes), and reuse of foundation (blue box).

**Figure 5 materials-13-00405-f005:**
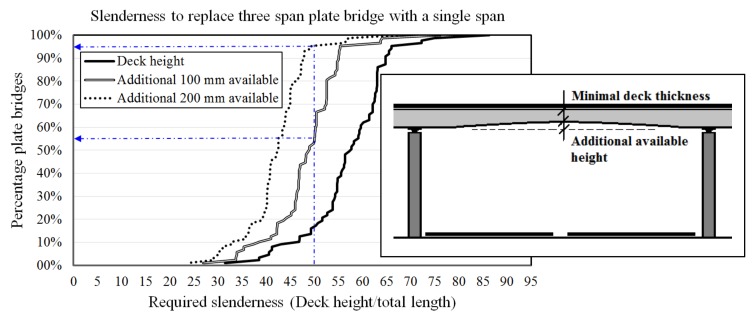
Required deck slenderness to realize a significant replacement task: processed data from Reference [11].

**Figure 6 materials-13-00405-f006:**
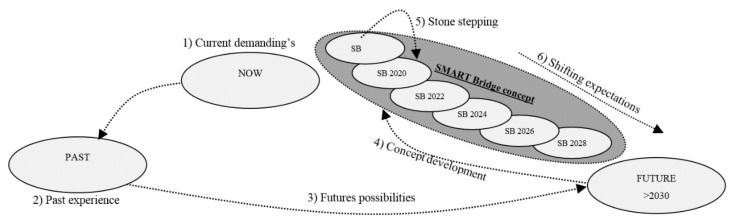
SMART bridge concept development process [13].

**Figure 7 materials-13-00405-f007:**
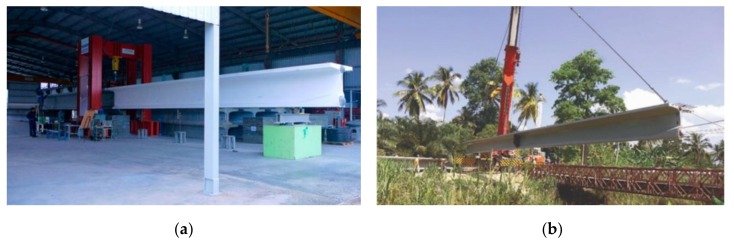
(**a**) Full-scale destructive test on a 30 m long segmental T-beam, research done in 2007–2010 [14,15]; (**b**) construction of 30-m-span Ulu Chemor bridge, stitched T-girder construction [15].

**Figure 8 materials-13-00405-f008:**
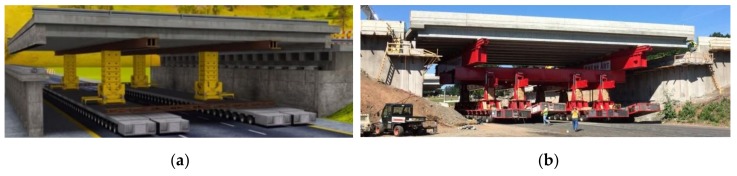
(**a**) Illustration of placing a new bridge deck with the use of an SPMT (computer-controlled platform vehicles) with jack-ups [19]; (**b**) placing a bridge with an SPMT during the replacement project, I-84 bridges over Marion Avenue in Southington, US [20].

**Figure 9 materials-13-00405-f009:**
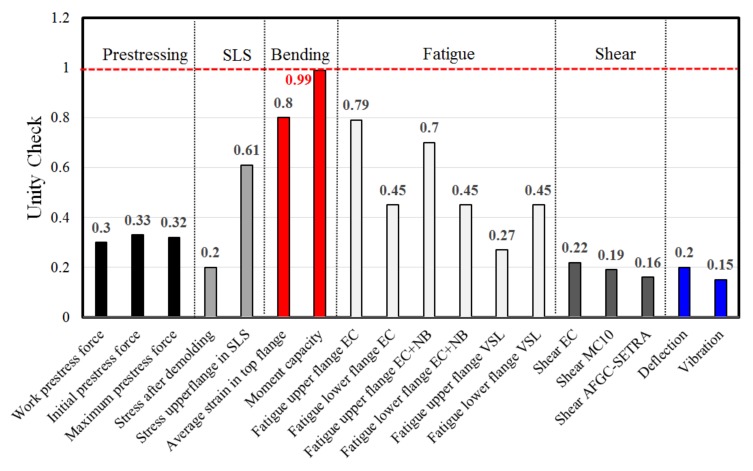
Outcome of the calculation with BoDeTo (ULS bending criteria is reached).

**Figure 10 materials-13-00405-f010:**
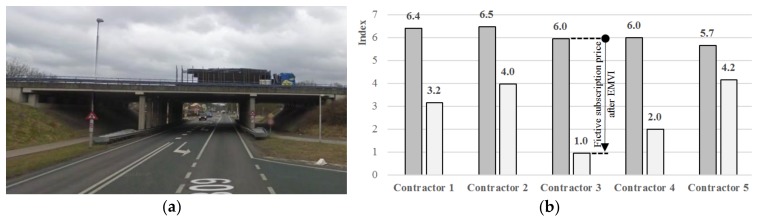
(**a**) Existing three-span plate bridge in the A28/N309; (**b**) Outcome of the tender with EMVI.

**Figure 11 materials-13-00405-f011:**
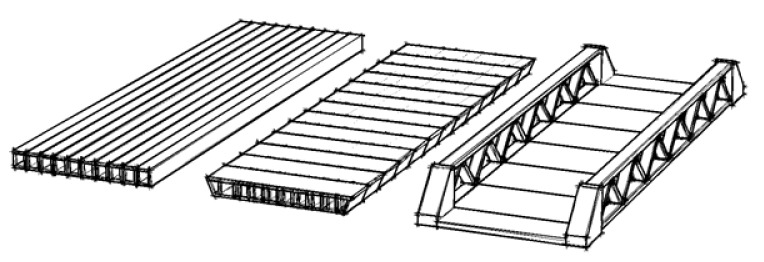
The three bridge types that are considered in the SMART bridge concept.

**Figure 12 materials-13-00405-f012:**
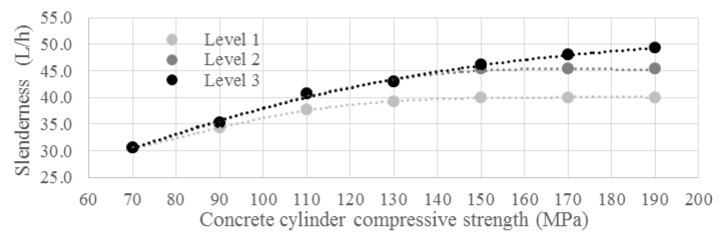
Achievable slenderness of a box girder bridge with an increasing concrete compressive strength.

**Figure 13 materials-13-00405-f013:**
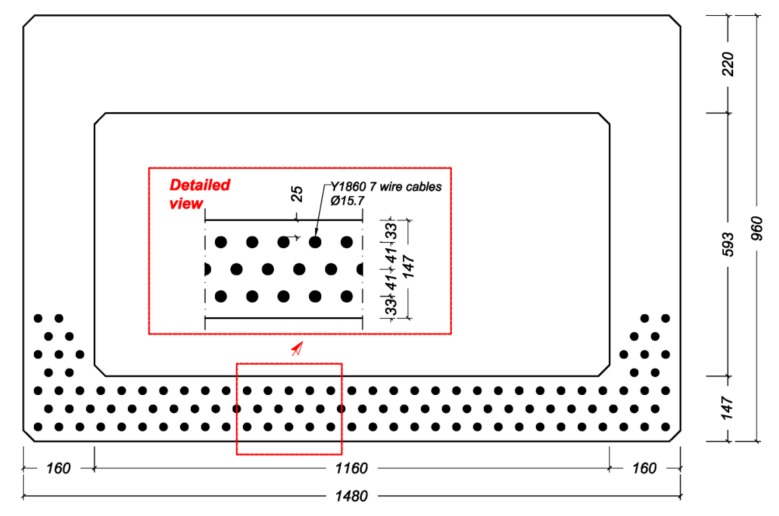
Designed cross-section of the girder with the distribution of the prestressing strands for the level 3 analysis (Figure 12).

**Figure 14 materials-13-00405-f014:**
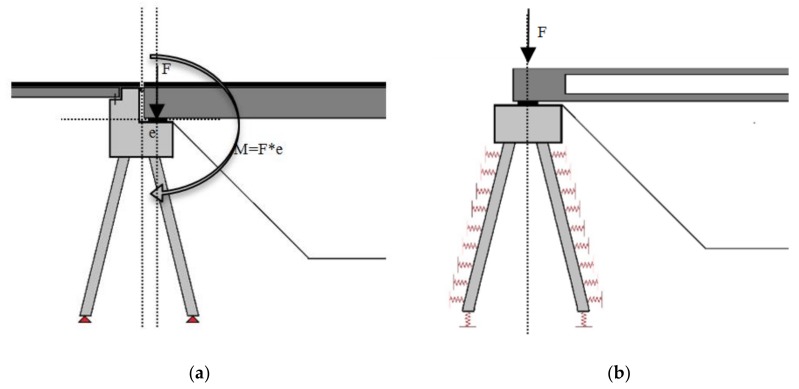
(**a**) Existing and (**b**) new bridge foundation design.

**Figure 15 materials-13-00405-f015:**
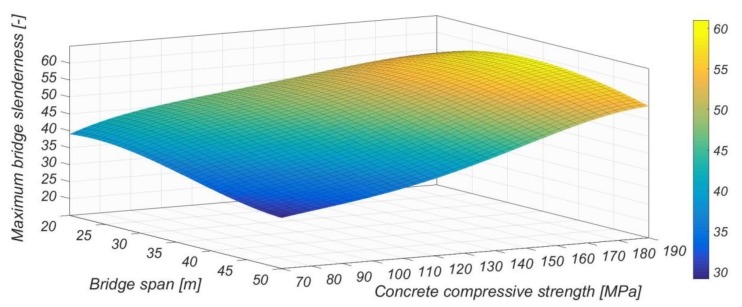
Achievable slenderness calculated with the BoDeTo.

**Figure 16 materials-13-00405-f016:**
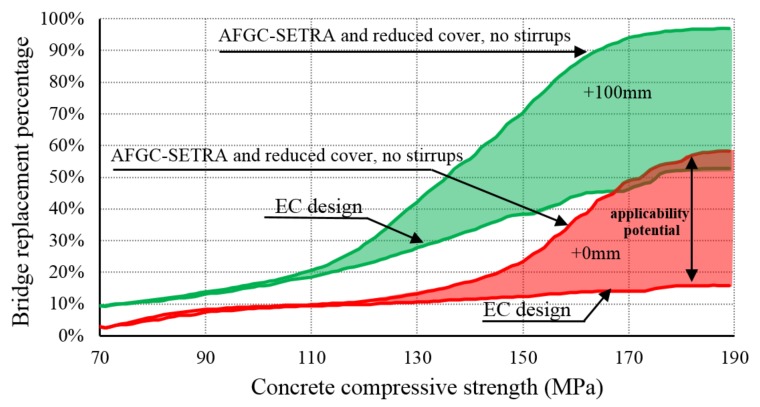
Percentage of three-span plate bridges in the Netherlands for which the SMART bridge replacement can be applied.

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
