# Peer review of "Future Infrastructural Replacement Through the Smart Bridge Concept"

_materials, 2020, doi:10.3390/ma13020405_

Round 1

Reviewer 1 Report

The Authors present a new strategy for replacing old highway bridges principally aimed at minimizing the traffic interruptions and the construction time. The proposed approach combines the use of advanced cementitious materials (in particular, UHPC), techniques for accelerated bridge construction, advanced design tools and the use of full-scale tests and of Structural Health Monitoring (SHM) techniques.

The subject is interesting and relevant for practical applications, and the paper is well written. I suggest the following minor revisions for improving the completeness and the impact of the paper.

1) The Introduction can be enriched by reporting the results of advanced studies on bridges having the same features of those examined. Moreover, the choices at the basis of the proposed concept have to be explained. For example, why only UHPC prestressed plates are considered, and not steel or mixed steel-concrete structures are considered? Why other kinds of structural schemes are ruled out? It is due to costs, or to technological traditions of Netherlands, or to other reasons?

2) About the SHM techniques, I think that the new fast and contactless remote technique of the radar interferometry can be fruitfully coupled with the proposed concept. On this matter, I suggest referring to the following two papers: a) DOI: 10.1109/EESMS.2017.8052699 and b) https://doi.org/10.1016/j.ndteint.2008.04.005.

Reviewer 2 Report

The advantage of this paper is a complex study of replacement of three span viaduct by one span with ambitious slenderness up to 50. Unfortunately there is no information about stress value in UHPC reached in design process. SLS conditions - deflections, dynamic response of the span under live load are not presented. Finally a creeping effect should be considered too.

Reviewer 3 Report

The subject of the paper is interesting, the authors should consider "major comprehensive" revisions to improve the manuscript. Please see remarks below.

1.Figure 3 does not bring anything new. I suggest to delete the figure 3.

2.Figure 4, the view of the bridge does not correspond to the description underneath,please correct it.

3.Figure 5 - explain what is the source of plotted curves

4.Figure 9 - could you please discuss the outcome

5.It would be interesting to include the obtained numerical values in the example presented in section 4.2. In the current situation it is not possible to comment the results.Please add some obtained values

6.The Authors explain: ”In Figure 15 (indicated by red color) it is shown that by designing with UHPC, according to the AFGC-SETRA recommendation [20], applying a reduced cover and taking into account fiber reinforcement and no stirrups, instead of requirements given by EC, the potential application area for bridge replacement increases from 16% to 58%..”

Could you please explain in detail if above comment concerns the Netherlands or France or another country, and what type of Bridges?

Round 2

Reviewer 3 Report

Most of the comments were clarified by the authors and appropriate changes were made to the text of the paper. The description of the figures has been supplemented and the list of references has been extended.

However, the response to comment:

“It would be interesting to include the obtained numerical values in the example presented in section 4.2. In the current situation it is not possible to comment the results”.

 is not satisfying.

The authors answer:

“We agree here with the reviewer. However, including results and proper comment on them demands too detailed analysis, which is already performed in one of the publications. Therefore, the reference to this publication is given”.

Also in the reference [12]:

Reitsema, A., Lukovic, M., Hordijk, D. Towards slender, innovative concrete structures for replacement of existing viaducts. Proceedings of fib Symposium 2016 Performance-based Approaches for Concrete 452 Structures. Cape Town.

there is no all responses to the comment of the reviewer.

In the paper [12] some of the information is in the Fig. 11: Combined pretensioned prestressing and post-tensioning in a prefabricated box girder.

In summary, it would be necessary for the readers of the paper to include the information about:

(1) the most unfavorable load combination included in the calculation and selection of strands.

(2) the final number and location of strands in the cross section of the girder and the cross section of the bridge with number of girders.

because this is the essence of the case study : “4.2. Case study replacement task A28/N309”. I suggest to add the figure describing the above information.
